# GeneAR: Autoregressive Gene-to-WSI Tile Synthesis via Causal MeanFlow

## Abstract

Understanding how transcriptomic programs shape tissue morphology remains a central challenge in computational pathology. Gene-to-WSI tile synthesis offers a principled generative framework to translate molecular profiles into histological images. However, most existing methods compress RNA-Seq into a single global embedding injected once at initialization, an oversimplified design that weakens transcriptomic signals and induces non-causal associations between gene expression and tissue morphology. We present GeneAR, an Autoregressive Gene-to-WSI model that reformulates synthesis as an iterative, coarse-to-fine generative process. At its core is a novel Causal MeanFlow module that reinforces transcriptome-informed guidance at multiple stages and mitigates non-causal factors through counterfactual-style interventions, thereby ensuring biological fidelity throughout the generative trajectory. Combined with a $\beta$-VAE for compact gene embeddings and a multi-scale vector quantizer for discrete morphology representation, GeneAR generates H&E-stained WSI tiles that are both visually realistic and transcriptomically faithful. Extensive experiments across five TCGA cancer benchmarks demonstrate consistent state-of-the-art performance, surpassing prior methods in both generative fidelity and downstream classification accuracy. All models and code will be released to facilitate reproducibility.

## 1 Introduction

A central question in computational pathology is how transcriptomic programs shape tissue morphology, and whether pathology images can be generated directly from RNA-Seq to probe this link (Coudray et al., 2018; Schmauch et al., 2020). Framing this as *Gene-to-WSI tile synthesis* is both scientifically meaningful and practically useful: it enables in silico experiments on how molecular perturbations manifest morphologically, provides privacy-preserving synthetic data for model development, and improves label/data efficiency in downstream tasks such as cancer classification or biomarker discovery (Ktena et al., 2024; Chang et al., 2023). Beyond augmentation, Gene–to–WSI generation offers a controlled testbed for multi-omics hypothesis generation and can boost statistical power where real images or labels are scarce or imbalanced (Chen et al., 2025).

Despite its promise, Gene-to-WSI tile synthesis remains methodologically underdeveloped. Existing solutions typically compress RNA-Seq profiles into low-dimensional embeddings that either drive one-shot generators (e.g., GAN variants) (Carrillo-Perez et al., 2023) or condition cascaded diffusion pipelines (Carrillo-Perez et al., 2025). While feasible, these strategies face structural limitations that undermine biological fidelity and scalability. First, collapsing the transcriptome into a static embedding injected only once causes *signal decay*, with molecular guidance fading as generation unfolds and images drifting toward superficial correlations rather than gene-driven morphology. Second, synthesizing tiles at a fixed resolution introduces *scale rigidity*,

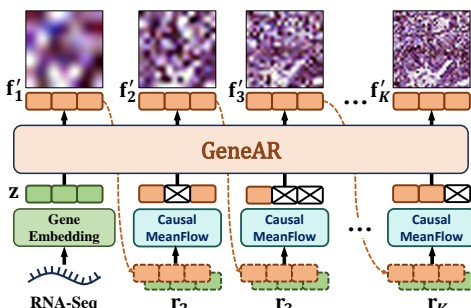

Figure 1: **Key Idea.** GeneAR integrates Causal MeanFlow into an autoregressive framework, where transcriptomic guidance is iteratively injected with causal enhancement to ensure biologically faithful WSI synthesis.

weakening cross-scale semantic consistency and diminishing transcriptome–morphology alignment. Third, embeddings are learned in a purely correlational manner, leaving models vulnerable to *confounders* such as batch effects, tumor purity, and staining variability, thereby undermining causal fidelity.

To address these challenges, we reformulate Gene-to-WSI synthesis as an *iterative, coarse-to-fine generative process* in which transcriptomic signals are injected at multiple stages during generation rather than used once at initialization. Building on this principle, we propose GeneAR—the first autoregressive Gene-to-WSI model that delivers stepwise molecular reinforcement while preserving structural coherence and causal fidelity (see Fig. 1). While drawing inspiration from recent advances in visual autoregression (Tian et al., 2024), GeneAR extends this paradigm into the molecular domain by conditioning coarse-to-fine prediction on compact, biologically grounded embeddings derived from a $\beta$-VAE. This integration ensures that the transcriptome remains an active driver of morphology, enabling dynamic cross-scale guidance during decoding. As a result, GeneAR prevents *signal decay*, enforces *multi-scale consistency*, and strengthens the causal alignment between gene programs and emergent tissue organization.

Yet an iterative paradigm alone does not immunize the generative trajectory against *confounders*—such as batch effects, tumor purity, and staining variability—whose correlational footprints can divert decoding from gene-driven semantics (Leek et al., 2010; Aran et al., 2015). To address this, Causal MeanFlow is introduced as a gene-driven causal module intrinsically embedded in the autoregressive trajectory. Inspired by average velocity fields (Geng et al., 2025), it reformulates flow dynamics in a biological context by directly coupling generative updates with transcriptomic embeddings. During training, counterfactual interventions disentangle causal signals from spurious variation, while at inference the learned field governs generation independently without auxiliary samples. More than an auxiliary add-on, Causal MeanFlow functions as an integral mechanism that injects transcriptomic guidance at multiple stages, suppresses non-causal drift, and preserves structural coherence. Through this integration, GeneAR establishes a generative framework in which transcriptomic signals remain active, consistent, and causally faithful throughout synthesis.

We validate GeneAR on five TCGA benchmarks spanning diverse cancer types. Across all datasets, GeneAR attains the lowest Fréchet Inception Distance (FID), indicating superior generative fidelity, and achieves the highest accuracy and F1 score in downstream cancer classification. These results confirm that GeneAR not only synthesizes realistic morphology faithfully aligned with transcriptomic profiles, but also produces synthetic data that directly enhance predictive modeling. Our main **contributions** are summarized as follows:

- **Paradigm Shift.** GeneAR reformulates Gene-to-WSI synthesis as an *iterative, coarse-to-fine generative process* in which transcriptomic signals are injected at multiple stages across scales, overcoming the signal decay and rigidity of static global embeddings.
- **Causal Fidelity.** By integrating the novel Causal MeanFlow module into the autoregressive trajectory, GeneAR disentangles transcriptomic effects from confounders, ensuring that morphological synthesis remains biologically grounded and causally aligned.
- **State-of-the-Art Performance.** Comprehensive evaluation on five TCGA cohorts demonstrates that GeneAR attains the lowest FID and the highest accuracy in downstream classification, setting a new benchmark for both generative fidelity and functional utility.

## 2 RELATED WORKS

**Gene-to-WSI Synthesis.** In recent years, most prior studies in computational pathology focus on predicting gene expression from WSIs (Schmauch et al., 2020; Pizurica et al., 2024; Chung et al., 2024; Ganguly et al., 2025), whereas the inverse task—synthesizing histology directly from RNA-Seq—remains underexplored. Early attempts such as RNA-GAN (Carrillo-Perez et al., 2023) inject transcriptomic information only once and operate at a fixed resolution, while cascaded diffusion models (Carrillo-Perez et al., 2025) improve visual fidelity but still rely on static conditioning and fail to account for confounders. Motivated by these challenges, we propose a *multi-stage, coarse-to-fine Gene-to-WSI framework* that injects transcriptomic guidance throughout decoding, preventing signal decay, and explicitly disentangles causal signals from spurious variation, thereby enabling more robust and biologically faithful histology synthesis.

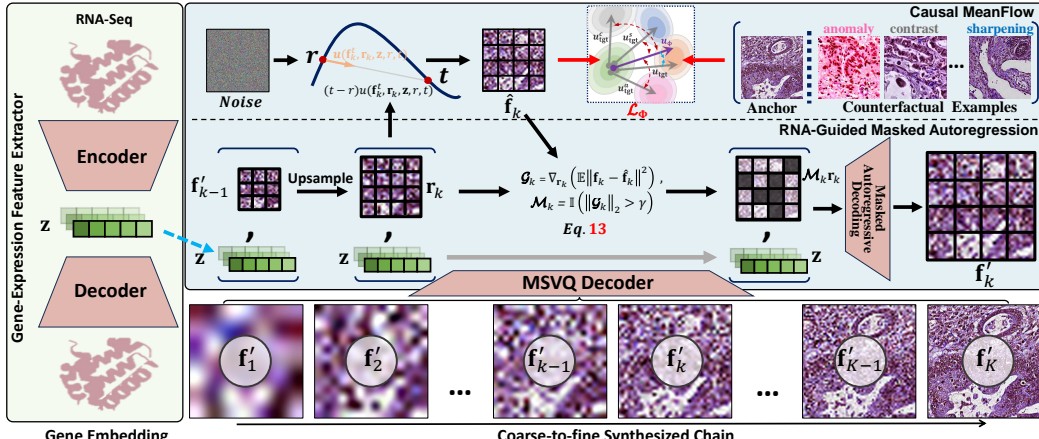

Figure 2: **Overview of GeneAR**. RNA-Seq embeddings are injected into a multi-scale autoregressive pipeline, where morphology is reconstructed from $\mathbf{f}'_K$ via the MSVQ decoder and reinforced by Causal MeanFlow with counterfactual supervision (see Sec. 3 for details).

**Autoregressive Image Generation.** Autoregressive transformers have reframed image synthesis as sequential token modeling (Van Den Oord et al., 2017; Razavi et al., 2019; Chen et al., 2020; Esser et al., 2021), further extended through residual quantization (Lee et al., 2022a), iterative refinement (Chang et al., 2022; Lee et al., 2022b), and scalable codebooks (Yu et al., 2023). Coarse-to-fine autoregression (Tian et al., 2024) achieves diffusion-level quality at lower sampling cost. Despite these advances, applications in computational pathology remain absent, where Gene-to-WSI pipelines still rely on GANs or diffusion (Carrillo-Perez et al., 2023; 2025). Our study introduces the first multi-scale autoregressive framework for RNA-guided WSI tile generation that preserves transcriptomic guidance and cross-scale consistency.

**Causal Generative Modeling.** Gene-to-WSI methods remain vulnerable to confounders such as staining or tumor purity (Leek et al., 2010; Macenko et al., 2009; Aran et al., 2015), with normalization or domain adaptation providing only partial remedies. In contrast, causal generative modeling (Schölkopf et al., 2021; Wu et al., 2024; Gao et al., 2025) and flow-based dynamics (Geng et al., 2025) offer new approaches to disentangling causal signals and improving stability, yet these advances have not been applied in pathology. We introduce a Causal MeanFlow module that embeds causal constraints into autoregressive training, ensuring persistent gene-driven semantics and suppressing spurious correlations.

## 3 METHODOLOGY

### 3.1 PRELIMINARIES

**Reformulation.** Recent studies show that transcriptomic programs are tightly coupled with tissue morphology and can even define layered cytoarchitectural organization in development (Schmauch et al., 2020; Qian et al., 2025). This suggests that Gene-to-WSI synthesis should preserve transcriptomic guidance dynamically across scales. However, existing methods collapse RNA-Seq into static embeddings injected once, leading to signal decay, scale rigidity, and spurious correlations. To overcome these issues, we reformulate Gene-to-WSI synthesis as a coarse-to-fine autoregressive process, where an image is quantized into hierarchical token maps $\mathcal{S} = \{s_k\}_{k=1}^n$ and generated sequentially under recurrent guidance from $\mathbf{g}$:

$$s_k = \mathcal{P}_\Theta(s_{<k}, \mathbf{g}). \tag{1}$$

Here, $\mathbf{g}$ is injected at multiple stages rather than only once at initialization, ensuring that transcriptomic signals remain persistently active throughout the generative trajectory while leveraging the distributional strength of autoregressive models (Esser et al., 2021; Lee et al., 2022a; Tian et al., 2024).

**Multi-Scale Vector Quantization.** We employ the multi-scale vector quantization (MSVQ) (Tian et al., 2024) to discretize a WSI tile $X \in \mathbb{R}^{H \times W \times C}$ into $K$ hierarchical token maps $\mathcal{S} = \{s_k\}_{k=1}^K$,

where each $s_k \in \mathbb{R}^{h_k \times w_k}$ denotes a discrete map at scale $k$. Unlike single-token quantization, MSVQ outputs entire token maps, which reduces inference cost and preserves cross-scale structure. For autoregressive modeling, the final-scale map $s_K$ is excluded, and the remaining maps are embedded by $\mathcal{W}$ and interpolated by $\mathcal{U}$, producing latent features $\mathcal{F} = \{\mathbf{f}_k\}_{k=1}^K$ and token embeddings $\mathcal{R} = \{\mathbf{r}_k\}_{k=2}^K$:

$$\mathbf{f}_k = \sum_{i=1}^k \mathcal{U}(\mathcal{W}(s_i), h_K \times w_K), \quad \mathbf{r}_{k+1} = \mathcal{U}(\mathbf{f}_k, h_{k+1} \times w_{k+1}), \tag{2}$$

where $\mathbf{f}_k \in \mathbb{R}^{h_K \times w_K \times d}$ aggregates multi-scale information up to level $k$, and $\mathbf{r}_{k+1} \in \mathbb{R}^{h_{k+1} \times w_{k+1} \times d}$ provides scale-aligned embeddings.

During decoding, similar to (Li et al., 2024b), the MSVQ decoder progressively reconstructs the tissue tile from $\mathbf{f}_K$ by residual refinement across scales, combining quantized maps and convolutional upsampling to recover both semantic structure and fine-grained details.

## 3.2 AUTOREGRESSIVE GENE-TO-WSI TILE MODELING

We propose GeneAR, which reformulates gene-conditioned pathology synthesis as an *iterative, coarse-to-fine generative process*. As shown in Fig. 2, GeneAR incorporates transcriptomic signals through three key components: **(1)** *Gene-Expression Feature Extractor*, which encodes RNA-Seq profiles $\mathbf{g} \in \mathbb{R}^{17655}$ into a compact latent prior $\mathbf{z} \in \mathbb{R}^{200}$; **(2)** *Causal MeanFlow*, which refines scale-wise responses $\mathbf{r}_k$ into causality-enhanced features $\hat{\mathbf{f}}_k$ under the guidance of $\mathbf{z}$; **(3)** *RNA-guided Masked Autoregression*, which constructs masks $\mathcal{M}_k$ from discrepancies between $\hat{\mathbf{f}}_k$ and $\mathbf{f}_k$ to highlight transcriptomic cues, and concatenates $\mathbf{z}$ with $\{\mathcal{M}_k \mathbf{r}_k\}_{k=2}^K$ for autoregressive decoding into the reconstructed token maps $\hat{\mathcal{S}} = \{\mathbf{s}_k'\}_{k=1}^K$, together with features $\hat{\mathcal{F}} = \{\mathbf{f}_k'\}_{k=1}^K$.

**Gene-Expression Feature Extractor.** Transcriptomic (RNA-Seq) profiles are inherently high-dimensional, necessitating compact yet biologically meaningful representations for effective integration into generative models. Following RNA-CDM (Carrillo-Perez et al., 2025), we adopt a $\beta$-VAE (Higgins et al., 2017) with a two-layer encoder $f_\psi$ and decoder $g_\theta$. Each RNA-Seq profile $\mathbf{g}$ is projected into a latent code $\mathbf{z}$, from which a reconstruction $\mathbf{g}'$ is generated. The training objective is

$$\mathcal{L}_{\psi,\theta} = -\mathbb{E}_{f_\psi(\mathbf{z}|\mathbf{g})}[log(g_\theta(\mathbf{g}|\mathbf{z}))] + \beta \cdot \mathbb{KL}(f_\psi(\mathbf{z}|\mathbf{g})||p(\mathbf{z})), \tag{3}$$

with $\beta$ regulating the trade-off between reconstruction fidelity and latent regularization. The resulting $\mathbf{z}$ serves as a compact molecular prior that conditions downstream generative modeling.

**Causal MeanFlow.** At higher scales, token maps expand rapidly; many positions become trivially predictable from earlier steps, inducing redundancy that over-smooths attention and erodes morphological fidelity (Guo et al., 2025). Moreover, conventional autoregressive models lack causal grounding, resulting in spurious correlations and weakened transcriptomic conditioning. To overcome these limitations, we propose Causal MeanFlow (r-$\mathcal{CM}$), an RNA-conditioned module that enforces causality and preserves gene-driven guidance across scales.

**a.) RNA-conditioned Average Velocity Modeling.** Departing from classical flow matching based on instantaneous velocity $v$, we adopt an average-velocity formulation, inspired by Geng et al. (2025), where $u$ is defined as the displacement between two time steps normalized by their interval. This design reduces multi-step integration to a single step, while the instantaneous velocity $v$ can be derived from the interpolated latent $\mathbf{f}_k^t$ at time $t$ and a noise sample $\epsilon_k$, expressed as:

$$\mathbf{f}_k^t = t\mathbf{f}_k + (1-t)\epsilon_k, \ \epsilon_k \sim \mathcal{N}(0,1),$$
$$v(\mathbf{f}_k^t, t) = \frac{d(\mathbf{f}_k^t)}{dt} = \mathbf{f}_k - \epsilon_k. \tag{4}$$

By exploiting the correlation between the average velocity $u$ and the instantaneous velocity $v$, differentiation with respect to $t$ yields a closed-form expression for $u$:

$$(t-r)u(\mathbf{f}_k^t, \mathbf{r}_k, \mathbf{z}, t, r) = \int_r^t v(\mathbf{f}_k^\tau, \tau) \, d\tau, \tag{5}$$

$$u(\mathbf{f}_k^t, \mathbf{r}_k, \mathbf{z}, t, r) = v(\mathbf{f}_k^t, t) - (t-r)(v(\mathbf{f}_k^t, t)\partial_{\mathbf{f}_k} u + \partial_t u). \tag{6}$$

We then employ a network $\Phi$, to estimate $u$ conditioned on $\mathbf{z}$ and $\mathbf{r}_k$, with the target average velocity $u_{\text{tgt}}$ expressed as:

$$u_{\text{tgt}} = v(\mathbf{f}_k^t, t) - (t - r)(v(\mathbf{f}_k^t, t)\partial_{\mathbf{f}_k} u_\phi + \partial_t u_\phi),$$
$$\text{where} \quad u_\phi = \Phi(\mathbf{f}_k^t, \mathbf{r}_k, \mathbf{z}, t, r). \tag{7}$$

Our network $\Phi$ extends beyond Geng et al. (2025) by coupling semantic features from $\mathbf{r}_k$ with transcriptomic signals from $\mathbf{z}$ in the prediction process. This fusion is implemented via biology-enhanced adaptive layer normalization block (adaLN) (Peebles & Xie, 2023), enabling position-wise integration of semantic and biological cues. The procedure is expressed as follows.

$$\alpha_1, \beta_1, \gamma_1 = \mathbf{mlp}(\mathbf{z} + t + r); \ \alpha_2, \beta_2, \gamma_2 = \mathbf{mlp}(\mathbf{r}_k + t + r),$$
$$\tilde{\mathbf{f}} = \mathbf{f}_k^t + \mathbf{attn}(\beta_1 \odot \text{LN}(\mathbf{f}_k^t) + \gamma_1) \odot \alpha_1, \tag{8}$$
$$\dot{\mathbf{f}} = \mathbf{mlp}(\beta_2 \odot \text{LN}(\tilde{\mathbf{f}} + \mathbf{cattn}(\tilde{\mathbf{f}}, \mathbf{r}_k)) + \gamma_2) \odot \alpha_2,$$

where $\dot{\mathbf{f}}$ and $\tilde{\mathbf{f}}$ denote intermediate features in r-$\mathcal{CM}$, $\odot$ is the Hadamard product, and "attn," "cattn," and "LN" correspond to attention, cross-attention, and layer normalization, respectively. At inference, r-$\mathcal{CM}$ performs one-step sampling to reconstruct $\hat{\mathbf{f}}_k$ from the scale-aligned embedding $\mathbf{r}_k$:

$$\hat{\mathbf{f}}_k = \epsilon_k - \Phi(\epsilon_k, \mathbf{r}_k, \mathbf{z}, 1, 0), \ \epsilon_k \sim \mathcal{N}(0, 1). \tag{9}$$

**b.) Counterfactual Regularization.** Autoregressive modeling can be viewed as an iterative reconstruction of quantized features, producing latent representations $\{\mathbf{f}_k'\}_{k=1}^K$. Within our framework, $\mathbf{f}_k'$ is reconstructed from $\mathbf{r}_k$ under the guidance of gene embedding $\mathbf{z}$, progressively enriching semantics across scales. To prevent spurious enhancements from non-causal factors such as color, contrast, or local frequency (Macenko et al., 2009; Tellez et al., 2019; Leek et al., 2010; Aran et al., 2015), we introduce a causal regularization strategy based on counterfactual interventions. Following the principle that interventions must destruct non-causal cues (Zhang et al., 2025), we apply three perturbations—color anomaly (to model stain variation), contrast adjustment (to capture batch effects and scanner differences), and sharpening (to simulate high-frequency artifacts) to generate counterfactual variants $X^a, X^c, X^s$ for each WSI tile $X$. After quantization, these yield feature maps $\{\mathbf{f}_k^a, \mathbf{f}_k^c, \mathbf{f}_k^s\}_{k=1}^K$, which force $\Phi$ to emphasize causal, scale-invariant morphology.

To endow r-$\mathcal{CM}$ with causal regularization, we enlarge the discrepancy between the predicted average velocity $u_\phi$ and the target velocities $\{u_{\text{tgt}}^a, u_{\text{tgt}}^c, u_{\text{tgt}}^s\}$ obtained from degraded counterfactuals, thereby stabilizing reconstruction. For example, $u_{\text{tgt}}^a$ is not derived via costly partial derivatives in Eq. 8. Instead, we decompose average velocity into magnitude and direction, where the direction is computed by normalizing the flow between two sampled fields $\mathbf{f}_k^{a,t}$ and $\mathbf{f}_k^{a,r}$, and the magnitude is estimated by scaling $||u_{\text{tgt}}||$ with a stochastic factor $\lambda$. Targets $\{u_{\text{tgt}}^c, u_{\text{tgt}}^s\}$ are constructed analogously. The procedure is formally expressed as:

$$u_{\text{tgt}}^a = \lambda_a ||u_{\text{tgt}}|| \cdot \frac{\mathbf{f}_k^{a,t} - \mathbf{f}_k^{a,r}}{||\mathbf{f}_k^{a,t} - \mathbf{f}_k^{a,r}||}. \tag{10}$$

**c.) Learning Objective.** We further define a causality-driven learning objective to disentangle scale-invariant morphological semantics (causal factors) from degradation-related cues (non-causal factors). For the $k$-th iteration, $u_{\text{tgt}}$ from $\mathbf{f}_k$ serves as the anchor, while $\mathbf{f}_{k-1}$ is treated as an extreme counterfactual, yielding $u_{\text{tgt}}^l$ via Eq. 10. Additional degradation samples $\{u_{\text{tgt}}^a, u_{\text{tgt}}^c, u_{\text{tgt}}^s\}$ derived from independent WSIs, distinct from the source of $u_{\text{tgt}}$, forming the counterfactual set $\mathcal{C}_u$. This design blocks potential shortcuts that exploit non-causal signals and enforces reliance on scale-invariant morphology. The training objective is:

$$\mathcal{L}_\Phi = \mathbb{E}\big[||u_\phi - \text{sg}(u_{\text{tgt}})||^2\big] - \frac{\alpha}{N} \sum_{u_{\text{tgt}}^n \sim \mathcal{C}_u} \mathbb{E}\big[||u_\phi - \text{sg}(u_{\text{tgt}}^n)||^2\big], \tag{11}$$

where $\text{sg}(\cdot)$ denotes stop-gradient and $\alpha$ controls the strength of causal regularization.

**RNA-Guided Masked Autoregression.** We formulate GeneAR as an autoregressive model over tokenized sequences:

$$p(s_1, s_2, \cdots, s_K) = \prod_{k=1}^K p(s_k | s_{<k}, s_0 = \mathbf{z}), \tag{12}$$

Table 1: **Quantitative comparison on TCGA benchmarks.** Fréchet Inception Distance (**FID, lower is better**) is reported along with model parameters (#Para) and generation steps (#Step). GeneAR consistently achieves the best performance across all cohorts.

| Type | Method | #Para | #Step | GBM | CESC | KIRP | COAD | LUAD | ALL |
|------|--------|-------|-------|-----|------|------|------|------|-----|
| Diff. | RNA-CDM (Carrillo-Perez et al., 2025) | 1146M | 2000 | 24.15 | 25.76 | 24.46 | 33.60 | 27.98 | 23.36 |
| | PathLDM (Yellapragada et al., 2024) | 400M | 50 | 22.46 | 19.87 | 20.39 | 26.96 | 21.28 | 20.29 |
| | U-ViT (Bao et al., 2023) | 297M | 100 | 13.89 | 15.74 | 21.83 | 26.75 | 17.86 | 18.55 |
| | DiT (Peebles & Xie, 2023) | 305M | 250 | 15.03 | 18.75 | 21.53 | 29.13 | 18.57 | 18.11 |
| FM | SiT (Ma et al., 2024) | 305M | 25 | 14.63 | 19.48 | 22.10 | 29.47 | 19.52 | 18.84 |
| AR | LlamaGen (Sun et al., 2024) | 343M | 256 | 15.48 | 18.34 | 19.89 | 27.91 | 17.52 | 17.43 |
| | VQGAN (Esser et al., 2021) | 227M | 256 | 21.74 | 23.48 | 26.10 | 32.47 | 26.48 | 25.09 |
| VAR | VAR (Tian et al., 2024) | 310M | 10 | 12.96 | 17.21 | 17.32 | 25.84 | 15.40 | 16.83 |
| | ImageFolder (Li et al., 2025) | 314M | 10 | 23.75 | 23.97 | 25.81 | 33.25 | 25.46 | 24.81 |
| | **GeneAR (Ours)** | 310M | 10 | **11.24** | **14.17** | **14.99** | **21.02** | **13.70** | **13.66** |

This factorization is realized by a ViT-based transformer with a causal mask, ensuring that generation proceeds strictly left-to-right under continuous transcriptomic conditioning, akin to GPT-2.

**a.) Gradient-Guided Masking.** After r-$\mathcal{CM}$ converges, it produces stable reconstructions $\hat{\mathbf{f}}_k$ from $\mathbf{r}_k$ before passing them to the transformer. We define the gradient as the derivative of the MSE loss between $\mathbf{f}_k$ and $\hat{\mathbf{f}}_k$ with respect to $\mathbf{r}_k$, and compute its $\ell_2$ norm along the channel dimension:

$$\mathcal{G}_k = \nabla_{\mathbf{r}_k}(\mathbb{E}||\mathbf{f}_k - \hat{\mathbf{f}}_k||^2), \;\; \mathcal{M}_k = \mathbb{I}(||\mathcal{G}_k||_2 > \gamma), \;\; \gamma \sim \mathcal{N}(0,1)^{h_k \times w_k}. \tag{13}$$

Here $\mathbb{I}(\cdot)$ denotes the indicator function. Tokens with large gradient responses are deemed RNA-conditioned salient regions, as perturbations there strongly affect reconstruction. These positions are therefore preferentially retained, dynamically injecting transcriptomic cues at the $k$-th iteration.

**b.) Masked Autoregressive Decoding.** Following class/text-conditioned VAR models, the latent gene embedding $\mathbf{z}$ is used as the initial condition token $\mathbf{r}_1$. For subsequent steps, tokens at positions with $\mathcal{M}_k = 0$ are replaced by a learnable embedding $\mathbf{e}$. The autoregressive generation and cross-entropy training objective are formulated as:

$$\{s_k^{'}\}_{k=1}^{K} = \mathcal{P}_\Theta(\{\mathbf{r}_1, \cdots, \mathcal{M}_k \mathbf{r}_{k-1}, \mathcal{M}_K \mathbf{r}_K\}), \;\; \mathcal{L}_\Theta = \sum_{k=1}^{K} \mathbb{CE}(s_k^{'}, s_k). \tag{14}$$

For efficiency, masking is applied only when $k \geq K_m$, balancing accuracy and cost. Together, these advances endow GeneAR with causality-aware, biologically grounded generation, which we validate in the next section through comprehensive experiments.

## 4 EXPERIMENTS

We evaluate GeneAR on five TCGA[1] cohorts following the RNA-CDM protocol (Carrillo-Perez et al., 2025). Each cohort comprises diagnostic WSIs paired with bulk RNA-Seq profiles and covers five cancer types with case counts: lung adenocarcinoma (LUAD, $n$=520), kidney renal papillary cell carcinoma (KIRP, $n$=298), colon adenocarcinoma (COAD, $n$=289), cervical squamous cell carcinoma and endocervical adenocarcinoma (CESC, $n$=277), and glioblastoma multiforme (GBM, $n$=212). Following (Li et al., 2021), we extract non-overlapping $256 \times 256$ tiles from gigapixel WSIs at $20 \times$ magnification; thus each RNA-Seq profile is associated with the set of tiles from its paired WSI. For generative quality, we report Fréchet Inception Distance (FID; 50K samples) (Heusel et al., 2017). For downstream evaluation, we train tile-level and WSI-level classifiers on synthetic tiles and report F1-score, accuracy (ACC), and AUC.

### 4.1 MAIN RESULTS

**Comparison with State-of-the-Art Methods.** We benchmark our GeneAR against nine strong baselines spanning four families—diffusion (Diff.), flow matching (FM), token-wise autoregres-

---

[1]The Cancer Genome Atlas Program

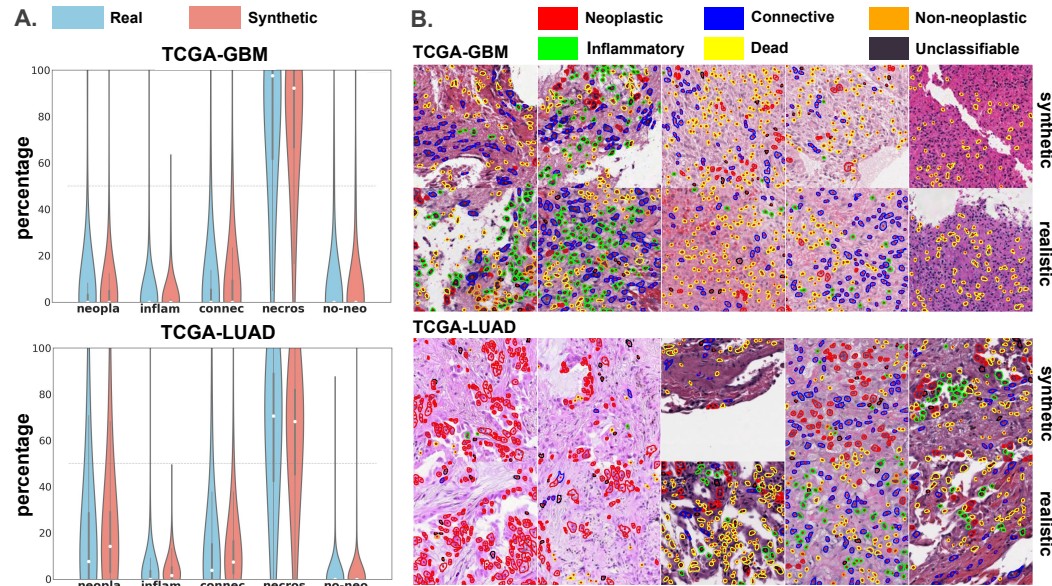

Figure 3: **Synthetic vs. real tiles**. *Panel A.* Cell-type distributions in real and synthetic tiles across TCGA cohorts. *Panel B.* HoverNet (Graham et al., 2019) visualizations showing consistent cellular composition between synthetic and real samples. Best viewed in color.

sion (AR), and scale-wise visual autoregression (VAR). All models are trained on the same cohorts under authors' recommended settings; when a class embedding is required, we replace it with an RNA-Seq embedding layer for parity. We report FID (50K, consisting of average samples spanning all RNA-Seq profiles) per cohort and on the pooled set (ALL). As shown in Tab. 1, our GeneAR achieves record-setting results on all five cohorts: GBM (11.24), CESC (14.17), KIRP (14.99), COAD (21.02), and LUAD (13.70). On the challenging COAD cohort, GeneAR improves over VAR (Tian et al., 2024) by 4.82 FID. Aggregated across cohorts, GeneAR achieves the best overall FID of 13.66 (ALL), outperforming LlamGen (Sun et al., 2024) by 3.77, DiT (Peebles & Xie, 2023) by 4.45, SiT (Ma et al., 2024) by 4.89, and RNA-CDM (Carrillo-Perez et al., 2025) by 9.70.

Table 2: **Cell distribution comparison.** Mean±std of cell proportions across cohorts. Extended results in Tab. 6 (Appendix).

| Dataset | Method | Neoplastic | Dead |
|---------|--------|------------|------|
| GBM | VAR | 9.96%±21.99 | 69.55%±34.59 |
| | **Ours** | **5.04%±12.07** | **78.67%±26.91** |
| | Real | 6.04%±16.29 | 77.49%±30.90 |
| LUAD | VAR | 25.06%±27.38 | 55.99%±30.00 |
| | **Ours** | **20.20%±22.78** | **61.10%±26.56** |
| | Real | 19.32%±25.21 | 63.12%±29.60 |

Table 3: **Tile classification performance.** ACC/F1 under varying substitution ($p$) and pretraining ($q$) ratios. Full results in Tab. 7 (Appendix).

| Method | $p=0.0$ | | $p=0.75$ | |
|--------|---------|---------|----------|---------|
| | ACC | F1 | ACC | F1 |
| R-CDM | 0.573±0.020 | 0.556±0.022 | 0.492±0.016 | 0.472±0.046 |
| VAR | 0.573±0.034 | 0.556±0.015 | 0.521±0.030 | 0.510±0.035 |
| **Ours** | **0.579±0.032** | **0.570±0.039** | **0.592±0.029** | **0.588±0.031** |
| | $q=0.5$ | | $q=1.0$ | |
| R-CDM | 0.618±0.026 | 0.612±0.030 | 0.650±0.021 | 0.641±0.031 |
| VAR | 0.661±0.020 | 0.662±0.019 | 0.708±0.011 | 0.709±0.010 |
| **Ours** | **0.722±0.023** | **0.722±0.020** | **0.767±0.015** | **0.766±0.017** |

**Cell Distribution Comparison.** While FID captures visual fidelity, biological plausibility is critical for clinical relevance. We therefore evaluate whether synthetic tiles preserve realistic cell composition using HoverNet (Graham et al., 2019), which segments and classifies cells into five categories: neoplastic, inflammatory, connective, dead, and non-neoplastic epithelial. For each benchmark, we randomly sample 2000 tiles from real WSIs and generate equal synthetic counterparts conditioned on RNA-Seq profiles with GeneAR. Quantitative results (Tab. 2) show that cell distributions in our GeneAR closely match real tissues and consistently outperform the strongest baseline VAR (Tian et al., 2024). For example, neoplastic cell proportions are 6.04%±16.29 vs. 5.04%±12.07 in GBM and 19.32%±25.21 vs. 20.20%±22.78 in LUAD. Visual comparisons in Fig. 3 further confirm that GeneAR not only preserves overall cell composition but also generates morphologies consistent with cancer type, demonstrating biological realism beyond mere visual fidelity.

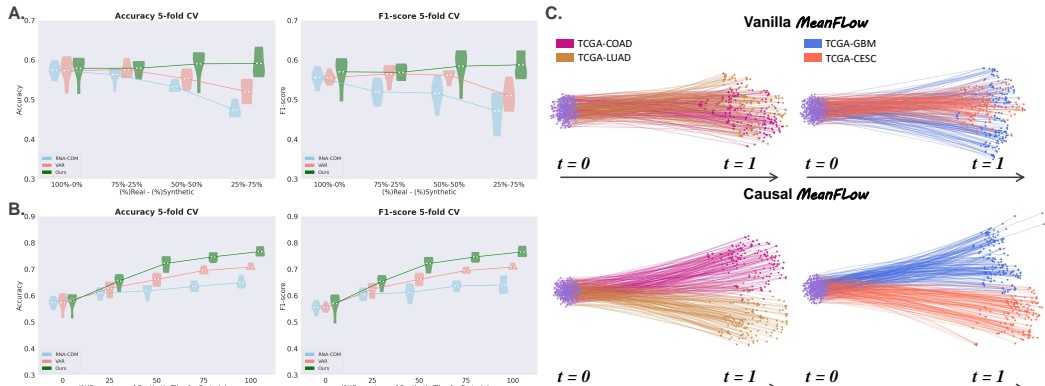

Figure 4: **Classification fidelity and causal dynamics.** *Panel A.* Conventional synthetic tiles degrade classification, whereas GeneAR preserves accuracy. *Panel B.* Pretraining with GeneAR-generated tiles yields the largest performance gains. *Panel C.* Causal MeanFlow disentangles trajectories, revealing class-specific dynamics absent in vanilla MeanFlow. Best viewed in color.

**Effect of Causal Regularization.** Fig. 4 C illustrates the impact of causal regularization on Mean-Flow learning. We employ t-SNE visualization, where points at $t = 0$ denote initial noise and those at $t = 1$ represent one-step generation outcomes; connecting lines approximate the average velocity $u$. Colors indicate different datasets. For vanilla MeanFlow (Geng et al., 2025), susceptibility to non-causal factors causes substantial overlap of velocity trajectories, reflecting poor class separability. In contrast, our Causal MeanFlow produces clearly separated trajectories, demonstrating its ability to emphasize scale-invariant morphological semantics (causal factors) and capture more fine-grained, class-specific histopathological features.

**Tile-Level Validation with Synthetic Data.** We used 5,000 model-generated tiles and 5,000 real tiles across five cancer types to evaluate whether synthetic data can sustain classification, employing a ResNet-18 (He et al., 2016) backbone for validation. Substituting up to 75% of real tiles with GeneAR-generated counterparts preserved or slightly improved performance (ACC: 0.579→0.592; F1: 0.570→0.588), whereas RNA-CDM (Carrillo-Perez et al., 2025) and VAR (Tian et al., 2024) caused notable drops (Tab. 3, Fig. 4 A). Pretraining with synthetic tiles further boosted performance, with GeneAR yielding the largest gains (+0.188 ACC, +0.196 F1) and outperforming baselines by clear margins (Tab. 3, Fig. 4 B). These results highlight the clinical utility of GeneAR, showing that model-generated tiles can serve as reliable training data.

**WSI-Level Gains with Synthetic Pretraining.** Extending to downstream WSI classification under the MIL paradigm, we benchmarked four state-of-the-art methods—TransMIL (Shao et al., 2021), ACMIL (Zhang et al., 2024), WiKG (Li et al., 2024a), and MambaMIL (Yang et al., 2024)—on COAD (MSS vs. MSI). As shown in Tab. 4, synthetic pretraining consistently improved all metrics, with ACMIL achieving the largest gains (ACC +0.096, F1 +0.109, AUC +0.121). These findings demonstrate that GeneAR-generated tiles not only substitute real data effectively but also enable highly discriminative slide-level representations, thereby enhancing downstream clinical classification tasks.

Table 4: **Performance comparison** of MIL methods (w/ vs. w/o pretraining). Our GeneAR generates 512 patch-wise tiles using each RNA-Seq.

| Method | ACC | | F1-score | | AUC | |
|---|---|---|---|---|---|---|
| | w/o | w/ | w/o | w/ | w/o | w/ |
| TransMIL | 0.849 | **0.877** | 0.847 | **0.875** | 0.894 | **0.934** |
| ACMIL | 0.767 | **0.863** | 0.749 | **0.858** | 0.817 | **0.938** |
| WiKG | 0.767 | **0.822** | 0.753 | **0.818** | 0.820 | **0.917** |
| MambaMIL | 0.836 | **0.918** | 0.833 | **0.917** | 0.925 | **0.941** |

### 4.2 ABLATION STUDY

**Analysis of RNA-conditioned Causal MeanFlow.** We assess the role of RNA-Seq conditioning and causal inference learning in r-$\mathcal{CM}$ under diverse settings. Reconstruction fidelity is measured by rFID, while FID quantifies overall generative quality within the autoregressive framework. A vanilla $\mathcal{MF}$ directly predicting from $\mathbf{r}_k$ serves as the control. As shown in Tab. 5a, incorporating causal inference learning reduces rFID to 3.70 compared with 4.41 when relying solely on RNA-Seq guidance, but yields a slightly higher FID (15.78 vs. 15.23). This indicates that while causal

Table 5: **Ablation experiments.** We systematically analyze (a) the components of r-$\mathcal{CM}$, (b) RNA-Seq embeddings, (c) core modules of GeneAR, (d) degradation strategies, (e) compatibility with different autoregressive models, and (f) masking strategies.

| Settings | rFID ($\downarrow$) | FID ($\downarrow$) |
|---|---|---|
| Vanilla $\mathcal{MF}$ | 6.63 | 16.05 |
| $\mathcal{MF}$ w/ RNA | 4.41 | 15.23 |
| $\mathcal{MF}$ w/ CI | 3.70 | 15.78 |
| r-$\mathcal{CM}$ | **2.03** | **13.66** |

(a) Key Components of r-$\mathcal{CM}$.

| Settings. | FID ($\downarrow$) |
|---|---|
| class-label emb. | 27.70 |
| linear emb. | 22.15 |
| $\beta$-VAE | **13.66** |

(b) RNA-Seq Embedding.

| Settings | | FID ($\downarrow$) |
|---|---|---|
| base.($\Delta$) | | 16.83 |
| | + ViT | 15.89 |
| $\Delta$ | + $\mathcal{FM}$ | 14.02 |
| | + r-$\mathcal{CM}$ | **13.66** |

(c) Key Components of GeneAR.

| $u_{\text{tgt}}^a$ | $u_{\text{tgt}}^c$ | $u_{\text{tgt}}^s$ | rFID ($\downarrow$) | FID ($\downarrow$) |
|---|---|---|---|---|
| ✔ | ✔ | | 2.25 | 14.36 |
| ✔ | | ✔ | 2.43 | 14.78 |
| | ✔ | ✔ | 2.78 | 15.01 |
| ✔ | ✔ | ✔ | **2.03** | **13.66** |

(d) Degradation Methods.

| Method | FID ($\downarrow$) |
|---|---|
| ImageFloder | 24.81 |
| + r-$\mathcal{CM}$ | 20.93 (3.88$\downarrow$) |
| VAR | 16.83 |
| + r-$\mathcal{CM}$ | 13.66 (3.17$\downarrow$) |

(e) Compatibility of r-$\mathcal{CM}$.

| Settings | FID ($\downarrow$) |
|---|---|
| w/o masking | 16.83 |
| Random | 16.26 |
| E-distance | 14.97 |
| Gradient | **13.66** |

(f) Masking Strategy.

learning enhances fidelity, RNA-Seq conditioning remains essential for optimal generative quality. Further analysis (Tab. 5d) on counterfactual sample combinations reveals consistent rFID gains of 2.16, 1.98, and 1.63 in rFID across different configurations relative to the baseline (4.41), with FID exhibiting a similar trend. Using all counterfactual variants achieves the lowest rFID and FID, confirming that the full set of degradation strategies is critical for effective causal regularization.

**Ablation for Components in GeneAR.** We first examine the role of RNA-Seq conditioning (Tab. 5b). Replacing RNA-Seq with class labels leads to a substantial FID degradation of 14.04, highlighting the necessity of biological modulation, while linear embeddings yield moderate improvements but remain 8.49 worse than the $\beta$-VAE, reflecting their limited ability to capture complex biological signals. At the model level (Tab. 5c), substituting r-$\mathcal{CM}$ with a non-generative ViT results in a 2.23 FID drop, indicating insufficient capacity to recover fine-grained morphology, whereas flow matching achieves a comparable FID of 14.02 but requires $25\times$ more steps, underscoring the efficiency advantage of r-$\mathcal{CM}$ in autoregressive modeling.

**Compatibility for Visual Autoregressive Modeling.** To assess scalability, we integrate r-$\mathcal{CM}$ with gradient-guided masking into two leading visual autoregressive frameworks, VAR (Tian et al., 2024) and ImageFolder (Li et al., 2025). Both benefit substantially, with FID gains of 3.17 for VAR and 3.88 for ImageFolder (Tab. 5e), demonstrating the broad compatibility and effectiveness of our design.

**Ablation for Masking Strategy.** We compare three masking strategies for selecting causality-enhanced tokens (Tab. 5f). Random masking slightly improves performance (+0.57 over baseline). Euclidean distance-based masking, which prioritizes poorly reconstructed tokens, further reduces FID by 1.86 relative to random masking, but fails to capture causal importance. In contrast, gradient-guided masking achieves the lowest FID of 13.66, outperforming Euclidean masking by 1.31, demonstrating its effectiveness in identifying causally salient tokens and enhancing generation quality.

## 5 CONCLUSION

In this work, we introduced GeneAR, an autoregressive Gene-to-WSI tile synthesis model that unites multi-stage transcriptomic conditioning with causality-aware modeling. By reformulating synthesis as an iterative, coarse-to-fine autoregressive process, GeneAR continually reinforces transcriptomic signals, mitigating signal decay and ensuring cross-scale consistency. At its core lies the Causal MeanFlow module, guided by RNA-Seq embeddings, which uses counterfactual interventions to suppress spurious variation and enforce causal fidelity in morphology. Experiments on five TCGA cohorts show that GeneAR achieves state-of-the-art generative fidelity and delivers consistent gains in downstream classification. Beyond benchmarking, GeneAR serves as a controllable tool for probing gene–morphology relationships under transcriptomic perturbations, opening new avenues for integrative studies in computational pathology.

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

## A APPENDIX

This appendix provides a comprehensive set of supplementary materials that extend and deepen the contributions presented in the main paper. We begin by revisiting the theoretical underpinnings of MeanFlow, offering a detailed derivation that clarifies its role in modeling average velocity and its integration into GeneAR. Next, we describe the construction of counterfactual samples and the implementation details of training, which together provide greater transparency and reproducibility of our method. Beyond methodology, we report expanded experimental results, including extensive ablation studies, complete quantitative evaluations, and large-scale comparisons with state-of-the-art baselines. Finally, we present supplementary qualitative visualizations that highlight the diversity and realism of the synthesized tiles across multiple cancer types. Taken together, these materials serve to reinforce the rigor of our approach, demonstrate its robustness across varied conditions, and provide deeper insights into its practical implications.

### A.1 FURTHER DERIVATION OF MEANFLOW

MeanFlow (Geng et al., 2025) introduces the concept of an average velocity $u$, defined as the displacement between two time steps $t$ and $r$, normalized by their interval. This reformulation reduces the multi-step integration of conventional flow matching to a single-step computation, thereby simplifying the generative process. Given Eq. 5, differentiating both sides with respect to $t$, we can rearrange this formulation to obtain $u$,

$$u(\mathbf{f}_k^t, \mathbf{r}_k, \mathbf{z}, t, r) = v(\mathbf{f}_k^t, t) - (t - r)\frac{d}{dt}u(\mathbf{f}_k^t, \mathbf{r}_k, \mathbf{z}, t, r). \tag{15}$$

where $\frac{d}{dt}u(\mathbf{f}_k^t, \mathbf{r}_k, \mathbf{z}, t, r)$ denotes the time derivative.

By further expanding the time derivative $\frac{d}{dt}u$ via the chain rule, $u$ can be explicitly expressed as:

$$\frac{d}{dt}u(\mathbf{f}_k^t, \mathbf{r}_k, \mathbf{z}, t, r) = \underbrace{\frac{d\mathbf{f}_k^t}{dt}}_{v(\mathbf{f}_k^t, t)}\partial_{\mathbf{f}_k}u + \underbrace{\frac{d\mathbf{r}_k}{dt}}_{0}\partial_{\mathbf{r}_k}u + \underbrace{\frac{d\mathbf{z}}{dt}}_{0}\partial_{\mathbf{z}}u + \underbrace{\frac{dt}{dt}}_{1}\partial_t u + \underbrace{\frac{dr}{dt}}_{0}\partial_r u,$$
$$= v(\mathbf{f}_k^t, t)\partial_{\mathbf{f}_k}u + \partial_t u. \tag{16}$$

In conclusion, the total derivative can be written as a Jacobian–vector product (JVP), where $[\partial_{\mathbf{f}_k}u, \partial_{\mathbf{r}_k}u, \partial_{\mathbf{z}}u, \partial_t u, \partial_r u]$ constitutes the Jacobian matrix of $u(\mathbf{f}_k^t, \mathbf{r}_k, \mathbf{z}, t, r)$ along the tangent vector $[v, 0, 0, 1, 0]$. Since the ground-truth function $u$ is inaccessible, we employ a network $\Phi$ to approximate it. Specifically, in Eq. 16, $u$ is replaced by $u_\phi$ learned via $\Phi$, and the target average velocity $u_{\text{tgt}}$ is ultimately given by:

$$u_{\text{tgt}} = v(\mathbf{f}_k^t, t) - (t - r)(v(\mathbf{f}_k^t, t)\partial_{\mathbf{f}_k}u_\phi + \partial_t u_\phi),$$
$$\text{where} \quad u_\phi = \Phi(\mathbf{f}_k^t, \mathbf{r}_k, \mathbf{z}, t, r). \tag{17}$$

Here, $\partial_{\mathbf{f}_k}u_\phi$ and $\partial_t u_\phi$ can be efficiently computed using the `jvp` interface in PyTorch. The procedure for training $\Phi$ with the target $u_{\text{tgt}}$ in Eq. 17 is summarized in Alg. 1.

### A.2 COUNTERFACTUAL SAMPLE CONSTRUCTION

Effective interventions should perturb non-causal factors while preserving causal content, thereby enabling meaningful causal inference. To this end, we empirically apply three interventions to the ground-truth tiles to construct counterfactual samples, as illustrated in Fig. 5.

**Color Anomaly.** Following CWNet (Zhang et al., 2025), we apply a color degradation procedure to the original tile $X$ in order to suppress color disturbances. The degradation is defined as:

$$X^a = \Delta H(X) + \Delta S(X) + \sum_{K \in \{R,G,B\}} \Delta K(X) + \epsilon, \tag{18}$$

where $H$, $S$, and $K$ denote the hue, saturation, and RGB channel offsets, respectively.

**Algorithm 1** Causal MeanFlow: Training

**Note:** `jvp` interface in PyTorch returns the function output and $\frac{du}{dt}$.

```
# fn(f_k, r_k, z, t, r): function to predict u
# f_k: gt quantinized feature at k scale
# r_k: token embeddings map at k scale
# z: compact molecular prior from RNA_Seq
# t,r: two sampled timestamps
# Counterfactual batch:
# f_k^l: r_k, f_k^a: color anomaly, f_k^c: contrast, f_k^s: sharpening

N = len([f_k^l, f_k^a, f_k^c, f_k^s])
t, r = sample_t_r()
ε = randn_like(f_k)

f_k^t = (1 - t) * f_k + t * ε
v_k^t = ε - f_k^t

u, du/dt = jvp(fn, (f_k, r_k, z, t, r), (v_k^t, 0, 0, 1, 0))
u_tgt = v_k^t - (t - r) * du/dt
u_tgt_m = ||u_tgt||

for f_k^j in {f_k^l, f_k^a, f_k^c, f_k^s}:
    ε_j = randn_like(f_k^j)
    f_k^{j,t} = (1-t) * f_k^j + t * ε_j
    f_k^{j,r} = (1-r) * f_k^r + r * ε_j
    d_j = f_k^{j,t} - f_k^{j,r}
    norm(d_j) = d_j / ||d_j||
    u_tgt^j = λ_j * u_tgt_m * norm(d_j)

C_u = {u_tgt^l, u_tgt^a, u_tgt^c, u_tgt^s}
error = u - sg(u_tgt)
for u_tgt^n in C_u:
    error_inven += (u - sg(u_tgt^n))

loss = metric(error) - α/N * metric(error_inven)
```

**Contrast Adjustment.** To prevent spurious semantic enhancement caused by contrast variation, we introduce a contrast adjustment intervention:

$$X^c = \sum_{K \in \{R,G,B\}} \left[ \alpha_K \cdot (X_K - \mu_K) + \mu_K \right] + \epsilon, \tag{19}$$

where $\alpha_K$ and $\mu_K$ are the contrast coefficient and channel-wise mean of the $K$-th channel.

**Sharpening Operation.** Similarly, to mitigate spurious semantic enhancement introduced by local spatial frequency distributions, we design a sharpening intervention:

$$X^s = X + \alpha \cdot (X - \text{Blur}(X)) \cdot k, \tag{20}$$

where $\alpha$ and $k$ control the sharpening intensity and lightness coefficient, respectively, and $\text{Blur}(\cdot)$ denotes Gaussian blurring.

## A.3 IMPLEMENTATION DETAILS

**MSVQ $\mathcal{Q}$.** All scales share a unified codebook $\mathcal{V} \in \mathbb{R}^{4096 \times 32}$, comprising 4096 entries of 32-dimensional vectors, with the number of discrete token maps per image fixed at $K = 10$. A DINOv2-based encoder (Oquab et al., 2023) is used to obtain continuous latent representations, followed by a decoder that reconstructs images from the quantized *token maps*. Subsequently, we

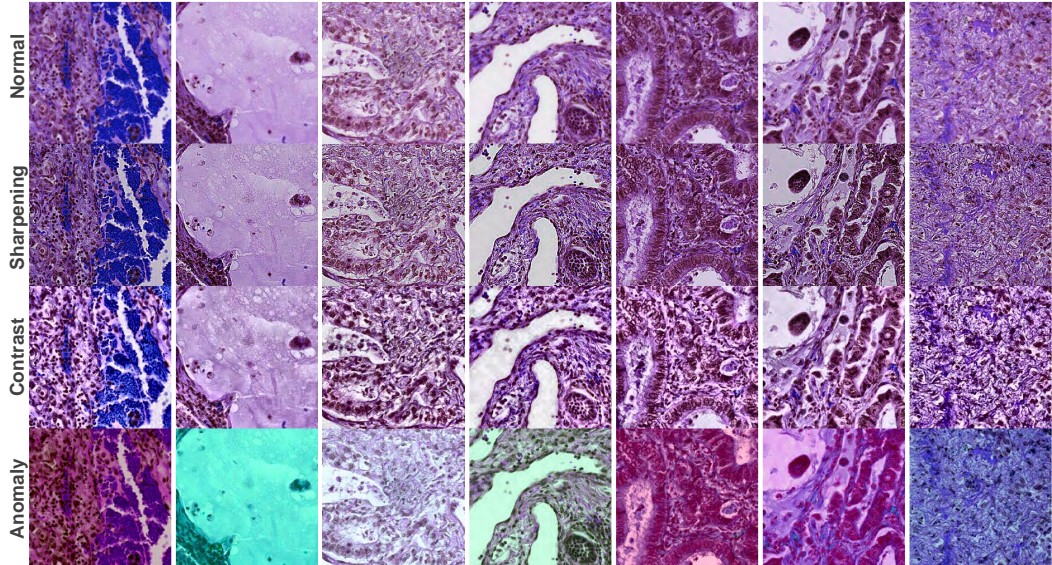

Figure 5: **Visualizations of counterfactual samples**. Our degradation procedures perturb non-causal factors while maintaining the morphology information.

randomly sample approximately 31 tiles per WSI associated with each RNA-Seq profile, constructing a total of 50,000 tiles, yielding an rFID of 2.11.

**Causal MeanFlow.** r-$\mathcal{CM}$ adopts the SiT architecture (Peebles & Xie, 2023) for the network $\Phi$, with model depth and feature dimension set to 6 and 768, respectively. The hyperparameter $\alpha$ is fixed at 0.1, and $\lambda$ is uniformly sampled from $[0.8, 1.2]$. r-$\mathcal{CM}_\Phi$ is trained for 100 epochs using 50 tiles per WSI. Additionally, we randomly sample approximately 31 tiles per WSI associated with each RNA-Seq profile, yielding a total of 50,000 tiles for rFID evaluation in Tab. 5a and Tab. 5d.

**RNA-Guided Masked Autoregression.** The VAR transformer follows the standard architecture with depth $d = 16$, head count $h = d$, and width $w = 64h$. Training is performed using an AdamW optimizer with initial learning rate $10^{-4}$, batch size 256, $\beta_1 = 0.9$, $\beta_2 = 0.95$, and weight decay 0.05. After pretraining r-$\mathcal{CM}$, we freeze its parameters and integrate it into the training of $\mathcal{P}_\Theta$ for 200 epochs using 200 tiles per WSI. The MSVQ module $\mathcal{Q}$ is trained jointly on the same tile sampling protocol.

**Code Availability.** The implementation of GeneAR, including training scripts and pretrained models, will be made publicly available upon publication.

## A.4 EXTENDED EXPERIMENTAL RESULTS

**Comprehensive Cell Distribution Comparison.** Tab. 6 reports the complete cell distributions of five cell types across five datasets, provided as a supplement to Tab. 2. For each dataset, we synthesize 2,000 tiles using all RNA-Seq profiles and randomly sample 2,000 real tiles. HoverNet (Graham et al., 2019), trained on the PanNuke dataset (Gamper et al., 2020), is employed to detect neoplastic, inflammatory, connective, dead, and non-neoplastic cell types. For each dataset, cell distributions are first quantified at the tile level, and then aggregated to obtain overall statistics across the 2,000 tiles. The results demonstrate that synthetic tiles generated by GeneAR better capture realistic cell-type distributions and further improve the morphological fidelity of synthesized tissue. Representative visualizations are provided in Fig. 6.

**Comprehensive Tile Classification Analysis.** Comprehensive results under different proportions $p$ and $q$ are reported in Tab. 7. For GeneAR, classification accuracy does not decline even as the fraction of synthetic tiles steadily increases within the 5000 mixed tiles, whereas all competing methods exhibit significant performance degradation. Moreover, classification accuracy continues to improve with the progressive incorporation of synthetic tiles during pretraining. Notably, GeneAR

Table 6: **Comparison of various cell distributions.** We systematically analyze the distributions of cell populations at the tile level across five cancer types, reporting both the mean and standard deviation (mean±std) of the corresponding cell proportion.

| Dataset | Method | Neoplastic | Inflammatory | Connective | Dead | Non-Neoplastic |
|---------|--------|-----------|--------------|------------|------|----------------|
| GBM | VAR | 9.96% ± 21.99 | 1.64% ± 5.46 | 15.87% ± 26.84 | 69.55% ± 34.59 | 2.95% ± 12.01 |
| | Ours | 5.04% ± 12.07 | 2.80% ± 7.60 | 8.97% ± 17.15 | 78.67% ± 26.91 | 4.49% ± 12.70 |
| | Real | 6.04% ± 16.29 | 2.82% ± 8.86 | 8.39% ± 19.19 | 77.49% ± 30.90 | 5.24% ± 16.06 |
| CESC | VAR | 30.92% ± 31.89 | 1.44% ± 5.20 | 19.04% ± 26.68 | 47.47% ± 32.18 | 1.10% ± 6.53 |
| | Ours | 27.19% ± 24.82 | 2.05% ± 5.52 | 13.31% ± 19.57 | 56.67% ± 26.64 | 0.77% ± 3.21 |
| | Real | 25.67% ± 28.43 | 2.73% ± 9.43 | 12.21% ± 21.66 | 58.77% ± 30.56 | 0.60% ± 2.95 |
| LUAD | VAR | 25.06% ± 27.38 | 2.99% ± 6.84 | 15.23% ± 20.63 | 55.99% ± 30.00 | 0.71% ± 3.42 |
| | Ours | 20.20% ± 22.78 | 3.95% ± 5.86 | 13.16% ± 15.14 | 61.10% ± 26.56 | 1.56% ± 6.64 |
| | Real | 19.32% ± 25.21 | 3.57% ± 6.86 | 12.46% ± 19.40 | 63.12% ± 29.60 | 1.52% ± 7.04 |
| KIRP | VAR | 20.59% ± 25.29 | 3.52% ± 8.01 | 9.22% ± 18.70 | 60.78% ± 26.97 | 2.72% ± 8.30 |
| | Ours | 22.61% ± 23.44 | 2.91% ± 4.64 | 9.36% ± 13.89 | 63.32% ± 25.53 | 1.78% ± 4.76 |
| | Real | 23.07% ± 26.71 | 2.44% ± 5.78 | 8.04% ± 16.01 | 64.43% ± 29.85 | 2.00% ± 7.23 |
| COAD | VAR | 33.37% ± 33.08 | 2.01% ± 7.29 | 27.39% ± 28.00 | 36.08% ± 29.55 | 1.13% ± 6.71 |
| | Ours | 38.14% ± 28.25 | 2.81% ± 5.00 | 20.09% ± 20.19 | 38.25% ± 24.69 | 0.69% ± 2.37 |
| | Real | 35.54% ± 32.95 | 3.32% ± 7.77 | 19.87% ± 24.85 | 40.49% ± 29.66 | 0.76% ± 3.45 |

Table 7: **Tile classification performance** under different substitution ratios $p$ of real tiles with synthetic ones, and varying proportions $q$ of synthetic tiles used for pretraining. Four different proportions for the pretraining 25% (1,250 samples), 50% (2,500 samples), 75% (3,750 samples) and 100% (5,000 samples). All experiments are conducted under a 5-fold cross-validation (CV) protocol to guarantee the stability and reliability of the results.

| Method | $p=0.0$ | | $p=0.25$ | | $p=0.50$ | | $p=0.75$ | |
|--------|---------|---------|----------|---------|----------|---------|----------|---------|
| | ACC | F1-score | ACC | F1-score | ACC | F1-score | ACC | F1-score |
| RNA-CDM | 0.573 ± 0.020 | 0.556 ± 0.022 | 0.563 ± 0.024 | 0.520 ± 0.028 | 0.533 ± 0.014 | 0.516 ± 0.036 | 0.492 ± 0.016 | 0.472 ± 0.046 |
| VAR | 0.573 ± 0.034 | 0.556 ± 0.015 | 0.576 ± 0.020 | 0.565 ± 0.026 | 0.553 ± 0.022 | 0.561 ± 0.017 | 0.521 ± 0.030 | 0.510 ± 0.035 |
| **Ours** | **0.579 ± 0.032** | **0.570 ± 0.039** | **0.580 ± 0.020** | **0.569 ± 0.020** | **0.590 ± 0.037** | **0.585 ± 0.042** | **0.592 ± 0.029** | **0.588 ± 0.031** |

| Method | $q=25\%$ | | $q=50\%$ | | $q=75\%$ | | $q=100\%$ | |
|--------|----------|---------|----------|---------|----------|---------|-----------|---------|
| RNA-CDM | 0.612 ± 0.021 | 0.606 ± 0.020 | 0.618 ± 0.026 | 0.612 ± 0.030 | 0.635 ± 0.017 | 0.637 ± 0.018 | 0.650 ± 0.021 | 0.641 ± 0.031 |
| VAR | 0.628 ± 0.025 | 0.625 ± 0.024 | 0.661 ± 0.020 | 0.662 ± 0.019 | 0.695 ± 0.013 | 0.698 ± 0.010 | 0.708 ± 0.011 | 0.709 ± 0.010 |
| **Ours** | **0.656 ± 0.023** | **0.654 ± 0.022** | **0.722 ± 0.023** | **0.722 ± 0.020** | **0.747 ± 0.016** | **0.747 ± 0.015** | **0.767 ± 0.015** | **0.766 ± 0.017** |

achieves larger gains than the two SOTA baselines. Classification models are trained for 50 epochs, with each pretraining stage fixed at 20 epochs.

**Comprehensive WSI Classification Analysis.** WSI classification follows the standard multi-instance learning framework, where all tiles extracted from an individual WSI form a tile-wise bag associated with a slide-level label: 0 for microsatellite stability (MSS) and 1 for microsatellite instability (MSI). Tile features are first extracted by a designated feature extractor and then fed into a classification model to predict the slide-level label. To assess the clinical utility of synthetic tiles, we adopt two tile feature extractors—ViT (Chen et al., 2021) pre-trained with MoCoV3 and CTransPath (Wang et al., 2022) pre-trained with SRCL—together with four state-of-the-art WSI classification models. We evaluate on the COAD MSI dataset (Kather et al., 2019), which includes 298 MSS and 66 MSI slides from 360 patients. Specifically, 292 slides for training and 72 slides for testing. For each RNA-Seq profile, GeneAR generates 512 synthetic tiles to construct tile-wise bags for classifier pretraining. As shown in Tab. 8, all classifiers consistently achieve significant gains across both feature extractors.

Table 8: **Performance comparison of MIL methods** with and without pretraining across four SOTA models. Shaded cells indicate results with pretraining.

| Method | ACC | | F1-score | | AUC | |
|---|---|---|---|---|---|---|
| | w/o | w/ | w/o | w/ | w/o | w/ |
| ViT pre-trained with MoCo V3 | | | | | | |
| TransMIL | 0.849 | 0.877 | 0.847 | 0.875 | 0.894 | 0.934 |
| ACMIL | 0.767 | 0.863 | 0.749 | 0.858 | 0.817 | 0.938 |
| WiKG | 0.767 | 0.822 | 0.753 | 0.818 | 0.820 | 0.917 |
| MambaMIL | 0.836 | 0.918 | 0.833 | 0.917 | 0.925 | 0.941 |
| CTransPath pre-trained with SRCL | | | | | | |
| TransMIL | 0.822 | 0.849 | 0.821 | 0.843 | 0.862 | 0.900 |
| ACMIL | 0.808 | 0.849 | 0.800 | 0.847 | 0.836 | 0.936 |
| WiKG | 0.767 | 0.836 | 0.745 | 0.829 | 0.876 | 0.912 |
| MambaMIL | 0.781 | 0.849 | 0.766 | 0.844 | 0.867 | 0.912 |

Table 9: **Ablation on the injection timing of r-$\mathcal{CM}$.** "Time" denotes the relative training time per epoch, while "Inject." indicates the number of r-$\mathcal{CM}$ injection operations. Shaded cell indicates the best FID.

| $K_m$ | FID $\downarrow$ | Time | Inject. |
|---|---|---|---|
| 7 | 14.39 | 1.51$\times$ | 4 |
| 8 | 13.52 | 1.30$\times$ | 3 |
| 9 | 13.66 | 1.00$\times$ | 2 |
| 10 | 15.74 | 0.86$\times$ | 1 |

## A.5 ADDITIONAL ABLATION RESULTS

**Ablation for $K_m$.** Injecting r-$\mathcal{CM}$ at every autoregressive step incurs considerable computational redundancy. To address this, we analyze token maps across scales $\{1, 2, 3, 4, 5, 6, 8, 10, 13, 16\}$ and find that the final two token maps account for 62.5% of the total sequence length (680). Based on this observation, the initial value of $K_m$ is empirically set to 9. We then evaluate GeneAR under a range of $K_m$ values. As reported in Tab. 9, setting $K_m = 8$ yields only a marginal improvement in FID (0.14) but comes at the cost of a 1.30$\times$ increase in per-epoch training time compared to $K_m = 9$.

## A.6 SUPPLEMENTARY VISUALIZATION RESULTS

To further illustrate the generative capacity of GeneAR, we provide additional qualitative results in Fig. 7. These synthetic H&E-stained tiles are generated across five representative cancer types (GBM, CESC, LUAD, KIRP, and COAD). The examples demonstrate that GeneAR is capable of producing diverse histological patterns that remain visually realistic while preserving disease-specific morphology. Such results highlight the robustness of GeneAR in synthesizing tissue structures that are consistent with biological priors, thereby complementing the quantitative evaluations reported in the main paper.

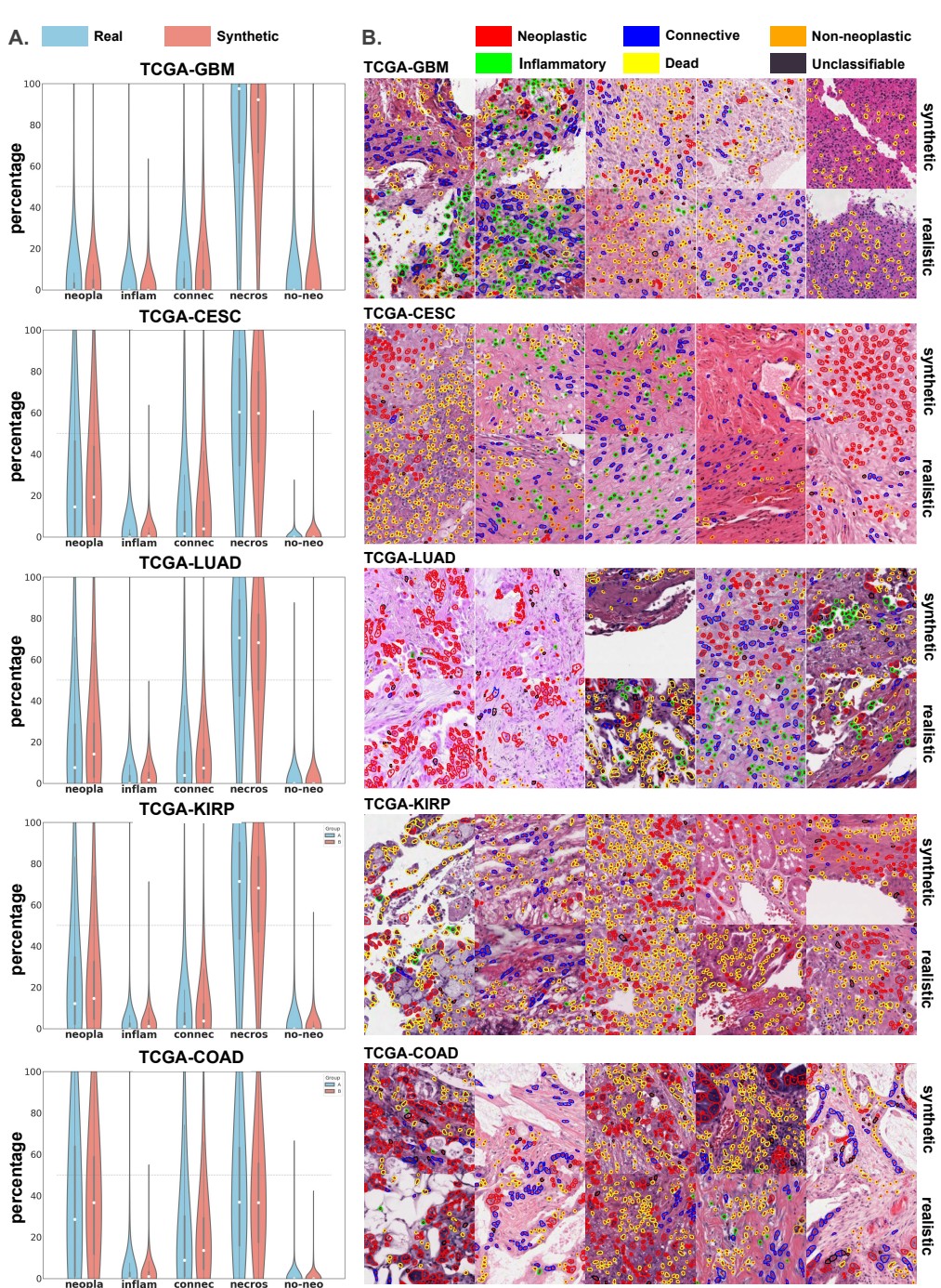

Figure 6: **Panel A.** Comparison of cell-type distributions between synthetic and real tiles across five cancer types. **Panel B.** Visualization of cells detected by HoverNet (Graham et al., 2019), showing neoplastic, inflammatory, connective, dead, and non-neoplastic populations in both synthetic and real samples.

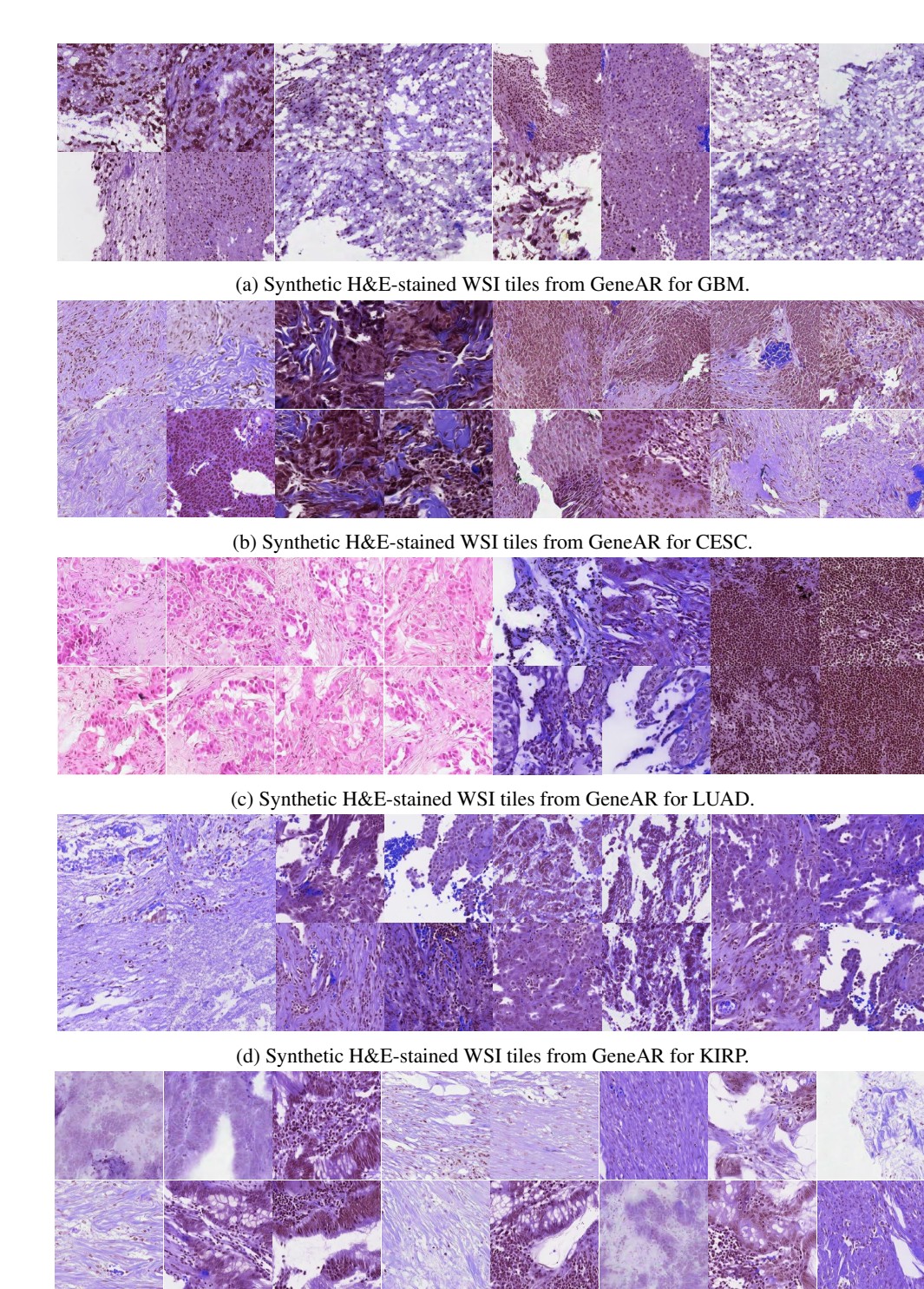

(a) Synthetic H&E-stained WSI tiles from GeneAR for GBM.

(b) Synthetic H&E-stained WSI tiles from GeneAR for CESC.

(c) Synthetic H&E-stained WSI tiles from GeneAR for LUAD.

(d) Synthetic H&E-stained WSI tiles from GeneAR for KIRP.

(e) Synthetic H&E-stained WSI tiles from GeneAR for COAD.

Figure 7: **Synthetic H&E-stained WSI tiles** generated by GeneAR across five cancer types: (a) GBM, (b) CESC, (c) LUAD, (d) KIRP, and (e) COAD. The results demonstrate that GeneAR can synthesize diverse and realistic histological patterns.

