# OpenReview forum: "GeneAR: Autoregressive Gene-to-WSI Tile Synthesis via Causal MeanFlow"
_ICLR.cc/2026/Conference — ICLR 2026 Conference Withdrawn Submission_

### Official Review · Reviewer_im4k · 2025-10-25

**Soundness:** 2
**Presentation:** 2
**Contribution:** 2
**Rating:** 2
**Confidence:** 5

**Summary:**

This method proposes an approach to synthesize HE images from RNA-seq. While the task is well-definied, this method lacks of details in methods as well as necessary benchmarking results.

**Strengths:**

The method utilizes causalflow-related method to generate HE images from expression profiles, and the method is well-definied, and the generated images look similar to real images.

**Weaknesses:**

This method lacks the details in method design, experiment setting, and the results interpretation.

1. Figure 2 is not very clear. In this case, RNA-seq works as conditions or sources? Which modeling approach is the best choice? Moreover, RNA-seq is a matrix, not a protein structure, so the authors need to replace the visualization approaches.

2. The motivation is not clear to me. Normally, performing sequencing+imaging is very expensive, and sequencing is also more expensive than imaging data. If we intend to generate an image based on expression profiles, that means we want to scale something from expensive to cheap, which does not make sense. Do authors have very strong motivations in predicting new modalities? For example, can any tasks be boosted when we have new image data?

3. This paper lacks important baselines included in the discussion and comparison. It seems that all of the benchmark studies in this work are ablation studies. The authors can investigate this survey https://arxiv.org/abs/2409.19365 and include more necessary SOTA baselines (3-5) to demonstrate the strength of their methods.

4. I do not see the hyperparameter settings, and the authors should report them.

5. Can we use this framework to generate images from spatial transcriptomic data? That will also be interesting. WSI image is always very highly-resolved, and I believe that the authors do not generate WSIs, so how to pair RNA-seq with the input images?

**Questions:**

Please see the weaknesses.

---

### Official Review · Reviewer_9RuP · 2025-10-28

**Soundness:** 2
**Presentation:** 3
**Contribution:** 2
**Rating:** 4
**Confidence:** 4

**Summary:**

GeneAR is an autoregressive framework translating RNA-Seq profiles into H&E histology tiles using a beta-VAE gene encoder, a Causal MeanFlow regularizer (image-space counterfactual perturbations), and a multi-scale vector-quantized transformer decoder.

**Strengths:**

1. Attempt at RNA to WSI synthesis is an ambitious and biologically meaningful problem.
2. The paper integrates causal-style regularization and multi-scale AR decoding.
3. It is Consistent (though small) quantitative gains through ablation within TCGA.

**Weaknesses:**

1. The paper repeatedly emphasizes causal fidelity and introduces a Causal MeanFlow, yet the notion of causality here is never rigorously defined. There’s no structural causal graph, no do-operator, and no evidence of true interventions on the underlying transcriptomic variables. Instead, the so-called counterfactuals are simply image-space degradations (color, contrast, sharpening) that may improve invariance but don’t demonstrate causal reasoning. If the authors genuinely aim to model gene-to-morphology causation, why not test morphological shifts after perturbing the expression of known driver genes or pathways? Without such experiments, the causal terminology risks being decorative rather than substantive.

2. All experiments are done on TCGA datasets that share preprocessing pipelines and scanner settings. If the method truly mitigates non-causal confounders like stain variation and batch effects, why not evaluate on a cross-institutional dataset such as CPTAC, PAIP, or even GTEx? Without an external test, it’s impossible to know whether the claimed robustness generalizes beyond the domain it was trained on. The five-cohort setup may look diverse on paper, but, in practice, it’s still one dataset in disguise.

3. The FID reductions, often 1–4 points, are small enough to fall within typical random-seed variance, and the downstream accuracy gains (0.579 → 0.592) are hardly convincing. No standard deviations, confidence intervals, or significance tests are reported. I find myself asking: Did the authors repeat the experiments? How stable are these numbers across runs or cohorts?

4. The authors correctly include recent diffusion (RNA-CDM, PathLDM, DiT), flow-matching (SiT), and autoregressive (LlamaGen, VAR) baselines. However, the comparison remains superficial: there is no discussion of training parity, compute fairness, or variance across runs.  Besides, why omit PixCell (https://arxiv.org/abs/2506.05127), a foundation model on histology synthesis benchmarks? Even if direct re-implementation was difficult, citing them and discussing relative compute or design trade-offs is expected. Moreover, evaluation is limited to FID, which is an image-space metric without biological validation or cross-dataset testing.

**Questions:**

1. How sensitive is the MeanFlow regularization to the α weighting coefficient and random λ factors? Was training stability or convergence affected? Are the results averaged across multiple seeds or single runs?

2. Since diffusion models are much larger (400–1000 M parameters) and trained with more steps, how do the authors ensure that the FID comparison is meaningful rather than simply reflecting compute disparity?

3. FID captures visual fidelity but not biological plausibility. Have the authors considered metrics assessing gene–morphology consistency, such as correlation of latent representations with expression clusters or pathway coherence?

4. Were FID results averaged across multiple seeds? What is the standard deviation across runs? Without variance reporting, how confident are the authors that the 1–3 FID gains are statistically significant?

5. If the model aims to learn gene-driven morphology, why not test causal consistency by perturbing the expression of known driver genes (e.g., TP53, KRAS) and observing whether generated morphology changes in expected ways?

---

### Official Review · Reviewer_ELyi · 2025-10-31

**Soundness:** 2
**Presentation:** 3
**Contribution:** 2
**Rating:** 6
**Confidence:** 2

**Summary:**

This paper introduces GeneAR, a gene-conditioned, multi-scale autoregressive generator for pathology tiles (H&E) guided by bulk RNA-Seq. Images are discretized with MSVQ into (K) token maps; features aggregated from previous scales (f_k) are upsampled into scale-aligned grids (r_{k+1}) that condition the next scale’s decoding. RNA profiles are compressed into a 200-D latent (z) by a β-VAE and injected throughout the pipeline. The technical core is Causal MeanFlow (r-CM) an average-velocity module that enables one-step dynamics and is trained with counterfactual regularization (color anomaly, contrast, sharpening) to suppress non-causal cues. On five TCGA cohorts (GBM, CESC, KIRP, COAD, LUAD) GeneAR reports state-of-the-art FID per cohort, and shows utility for downstream classification under synthetic-data substitution/pretraining.

**Strengths:**

1. Well-matched architecture to problem: Coarse-to-fine AR over MSVQ tokens with persistence of (z) and (r_k) provides continuous gene guidance across scales, more direct than single-shot conditioning.
2. r-CM is a thoughtful addition: Guiding AR with the counterfactual training is elegant. The objective is explicit and principled.
3. Solid empirical gains: GeneAR attains the best per-cohort FID, improving over strong diffusion, flow-matching, and AR/VAR baselines and gradient-guided masking is superior to random/Euclidean-distance. Downstream application results attained are also good.
4. Clear details: Writing is clear and can be followed. All the components are adequately explained.

**Weaknesses:**

1. "Causal” framing may overreach: The paper’s notion of causality is primarily enforced via engineered counterfactual degradations (stain, contrast, sharpening) and a push-pull loss. There is no explicit causal graph, identifiability result beyond these augmentations. The method reads as "regularized" MeanFlow with RNA conditioning - strong, but the “causal” claim would benefit from stronger evidence.
2. Biological faithfulness metrics are indirect: Authors rely heavily on FID and cell-type distributions (with HoverNet) is understandable but indirect for gene->morphology alignment. There’s no metric demonstrating how changes in RNA-seq can drive monotonic, localized morphological changes, given authors main claim to give conditioning in many steps is to maintain plausibility of morphologies.
3. Clarity on FID backbone: It is unclear how what backbone is used to calculate the FID. It is somewhat odd if standard Inception network is used in this case. People already have adapted foundation models in pathology for calculating FID metric, it would be more reliable to use FID calculated on such backbones.
4. No domain experts in the loop: Because of medical context, at least some samples should be evaluated by domain experts.

**Questions:**

1. Gene-image alignment: Is there any way this can be evaluated?
2. What exact causal effect are you claiming to identify (e.g., pathway activity to morphology)? How do you quantify this instead of using the fidelity metrics? Your “counterfactual” degradations are image-space perturbations. In what sense do these establish a biological causal relation? Is it just by removing shortcuts? If so there needs to be more evidence that removing shortcuts related to confounders does indeed lead to better causal generation of the images.
3. If FID is calculated on standard Inception type network, it would be less reliable, at any rate please clarify this in the manuscript.
4. Significance tests and evaluation from domain experts should increase confidence in the results presented.

---

### Official Review · Reviewer_97ju · 2025-11-01

**Soundness:** 3
**Presentation:** 3
**Contribution:** 3
**Rating:** 6
**Confidence:** 3

**Summary:**

This paper introduces GeneAR, a novel generative model for synthesizing histopathology (WSI) tiles from transcriptomic (RNA-Seq) profiles. The authors identify two key weaknesses in prior work (e.g., RNA-GAN, RNA-CDM) : (1) "signal decay" caused by compressing RNA-Seq into a single global embedding that is injected only once , and (2) vulnerability to non-causal confounders like staining variability and batch effects.

GeneAR proposes to solve this by reformulating synthesis as a coarse-to-fine autoregressive process , which allows for the iterative injection of transcriptomic guidance at multiple scales. The core of this mechanism is a novel "Causal MeanFlow" module, which is inspired by recent one-step flow models and uses counterfactual interventions to ensure causal fidelity. The authors validate GeneAR on five TCGA cohorts, demonstrating state-of-the-art results in both generative fidelity (FID) and, more importantly, downstream classification accuracy.

**Strengths:**

1. **High Architectural Novelty:** The paper's main strength is its sophisticated and novel generative architecture. The combination of a coarse-to-fine autoregressive framework (inspired by VAE modeling) to manage scale, with an efficient one-step flow model (inspired by MeanFlow) as the transition kernel, is a non-trivial and creative contribution.
2. **Valid Problem Formulation:** The authors correctly identify a significant limitation in the current SOTA models for this task. Prior works like RNA-GAN and RNA-CDM do rely on a static, "inject-once" $\beta$-VAE embedding. GeneAR's proposal of a multi-stage, iterative reinforcement of the transcriptomic signal is a logical and powerful solution to this "signal decay" problem.
3. **Strong Downstream Task Performance:** The authors are to be commended for prioritizing downstream classification accuracy over simpler pixel-level fidelity metrics. Demonstrating SOTA performance on this task shows that the synthetic images are not just visually realistic but also retain the functional, biologically-salient link between the gene expression profile and the morphological phenotype.

**Weaknesses:**

1. **Lack of Perturbation Validation:** The introduction explicitly frames the work as enabling "*in silico* experiments on how molecular perturbations manifest morphologically". This is the one exciting and scientifically significant application of such a model. However, the experimental section provides zero such case studies. This is a notable omission, as *in silico* perturbation is an active and established field of research (e.g., models like HistoXGAN , MoPaDi , and IMPA ).

2. **The "Causal" Claim is Unsubstantiated:** The paper claims that the model "mitigates non-causal factors" such as "batch effects" and "staining variability". This is an extraordinary claim that requires direct proof, but none is provided. The experimental design is missing:
   - (a) An ablation study comparing "Causal MeanFlow" to a standard (non-causal) "MeanFlow" to quantify the actual benefit of the counterfactual interventions.
   - (b) A direct validation of confounder robustness, such as a cross-hospital or cross-scanner generalization experiment, which is a standard method for testing robustness to batch effects. Without this evidence, the "Causal" descriptor is unsupported.

**Questions:**

1. Given the primary motivation of enabling *in silico* perturbation experiments , can the authors provide even a single case study demonstrating this capability? For example, can the model be used to generate counterfactual images showing the morphological effect of a specific known gene mutation, as has been demonstrated by other models in this domain?
2. What is the direct, quantitative evidence for the "Causal" claim? Can the authors please provide an ablation study or a cross-domain generalization experiment (e.g., train on N-1 hospitals, test on the held-out hospital) to prove that the "Causal MeanFlow" module measurably mitigates confounders like batch effects?

---

### Note · Authors · 2025-11-13

**Comment:**

Dear Area Chair and Reviewers,

We would like to sincerely thank all reviewers for their thoughtful and detailed feedback on our submission #7770, titled “GeneAR: Autoregressive Gene-to-WSI Tile Synthesis via Causal MeanFlow.” We deeply appreciate the time and effort invested in evaluating our work, as well as the constructive insights that helped us better understand both the strengths of the paper and the aspects that could be discussed further.

After careful consideration, we have decided to withdraw the paper from further consideration at ICLR 2026, as we believe that the current version may not be best positioned to fully address all points within the rebuttal phase. We plan to take more time to reflect on the valuable feedback and explore possible extensions before submitting a revised version to another venue.

Once again, we sincerely thank the reviewers and the area chair for their time, thoughtful assessments, and constructive suggestions.

Warm regards,
The authors of submission #7770

**Withdrawal Confirmation:**

I have read and agree with the venue's withdrawal policy on behalf of myself and my co-authors.